# Identification of nine new susceptibility loci for endometrial cancer

Tracy A. O'Mara [iD] et al.[#]

Endometrial cancer is the most commonly diagnosed cancer of the female reproductive tract in developed countries. Through genome-wide association studies (GWAS), we have previously identified eight risk loci for endometrial cancer. Here, we present an expanded meta-analysis of 12,906 endometrial cancer cases and 108,979 controls (including new genotype data for 5624 cases) and identify nine novel genome-wide significant loci, including a locus on 12q24.12 previously identified by meta-GWAS of endometrial and colorectal cancer. At five loci, expression quantitative trait locus (eQTL) analyses identify candidate causal genes; risk alleles at two of these loci associate with decreased expression of genes, which encode negative regulators of oncogenic signal transduction proteins (*SH2B3* (12q24.12) and *NF1* (17q11.2)). In summary, this study has doubled the number of known endometrial cancer risk loci and revealed candidate causal genes for future study.

Correspondence and requests for materials should be addressed to T.A.O'M. (email: Tracy.OMara@qimrberghofer.edu.au) or to A.B.S. (email: Amanda.Spurdle@qimrberghofer.edu.au) or to D.J.T. (email: djt25@medschl.cam.ac.uk). [#]A full list of authors and their affliations appears at the end of the paper.

Endometrial cancer accounts for ~7% of new cancer cases in women[1] and is the most common invasive gynecological cancer in developed countries (http://gco.iarc.fr/today/home). Risk of endometrial cancer is approximately double for women who have a first degree relative with endometrial cancer[2,3]. Rare high-risk pathogenic variants in mismatch-repair genes, *PTEN*, and DNA polymerase genes[4] explain a small proportion of endometrial cancers, and the eight previously published common endometrial cancer-associated single-nucleotide polymorphisms (SNPs) identified by genome-wide association studies (GWAS) studies[5–8] together explain <5% of the familial relative risk (FRR).

Here, we conduct a meta-GWAS including 12,906 endometrial cancer cases and 108,979 country-matched controls of European ancestry from 17 studies identified via the Endometrial Cancer Association Consortium (ECAC), the Epidemiology of Endometrial Cancer Consortium (E2C2) and the UK Biobank and report a further nine genome-wide significant endometrial cancer genetic risk regions. One of these risk regions on 12q24.12 was previously identified by meta-GWAS of endometrial and colorectal cancer[9]. eQTL and gene network analyses reveal candidate causal genes and pathways relevant for endometrial carcinogenesis.

## Results

**GWAS meta-analysis.** Details of genotyping for each study are found in Supplementary Data 1 and individual studies described in the Supplementary Information. Following standard quality control (QC) for each dataset (Supplementary Methods), genotypes were imputed using the 1000 Genomes Project v3 reference panel (combined with the UK10K reference panel for the WHI and UK Biobank studies). SNP-disease associations in each study were tested using logistic regression, adjusting for principal components, and risk estimates were combined using inverse-variance weighted fixed-effects meta-analysis. We found little evidence of genomic inflation in any dataset ($\lambda_{1000} = 0.996$–1.128) or overall ($\lambda_{1000} = 1.004$) (Supplementary Fig. 1). Using linkage disequilibrium (LD) score regression, we estimate that 93% of the observed test statistic inflation is due to polygenic signal, as opposed to population stratification.

Seven of the eight published genome-wide significant endometrial cancer loci were confirmed with increased significance (Table 1, Fig. 1a), although the effect sizes for some loci were slightly attenuated compared with our previous analysis (comprising 7737 cases and 37,144 controls[7], all also included in the current analysis). For example, the most significant SNP in this meta-analysis, rs11263761 intronic in *HNF1B*, had an odds ratio (OR) = 1.15 (1.12–1.19; $P = 3.2 \times 10^{-20}$), compared with OR = 1.20 (1.15–1.25; $P = 2.8 \times 10^{-19}$) in our previous analysis[7]. The previously reported associations with intronic *AKT1* SNPs (rs2498796 OR = 1.17 (1.07–1.17); $P = 3.6 \times 10^{-8}$)[6,10] did not reach genome-wide significance (rs2498796 OR = 1.07 (1.03–1.11) $P = 6.3 \times 10^{-5}$, Bayes false discovery probability (BFDP) 98%) in this meta-analysis, although the risk estimate direction is consistent with our original finding.

Excluding the 500 kb, either side of the risk loci previously reported at genome-wide significance for endometrial cancer alone, we found 125 SNPs with $P < 5 \times 10^{-8}$. Using approximate conditional association testing with GCTA software[11], these were resolved into nine independent risk loci; eight newly reported regions, plus the 12q24.12 locus previously identified by a joint endometrial-colorectal cancer analysis[9] (Table 2, Fig. 1b, Fig. 2a–i). The BFDP was ≤4% for all nine novel loci. The analysis was repeated with the restricted set of 8758 cases with endometrioid cancer, the most common histology (Fig. 1c); this identified one additional variant at 7p14.3 reaching genome-wide significance (rs9639594; Supplementary Data 2). However, given the sparse LD at this region and the fact that this is a single, imputed variant, further investigation of this region is required to confirm its association with endometrial cancer risk. No SNP reached genome-wide significance in an analysis restricted to the 1230 non-endometrioid cases (Fig. 1d) or in separate analyses of carcinosarcomas, serous, clear cell or mucinous carcinomas, for which statistical power is very limited (Supplementary Data 2, Supplementary Fig. 2).

For these nine newly reported endometrial cancer loci, a statistically significant difference in risk estimates by histological subgroup was observed only for the 2p16.1 locus; the risk was higher for non-endometrioid than for endometrioid cancer (rs148261157 OR = 1.64 (1.32–2.04) and OR = 1.25 (1.14–1.38), respectively, case-only $P_f = 0.003$, Table 2).

**Table 1 Meta-analysis results for previously identified genome-wide significant endometrial cancer risk loci**

| Region | SNP | Position (bp)[a] | Nearby gene(s) | Effect: other alleles | EAF | Info | All histologies (12,906 cases; 108,979 controls) | | | | Endometrioid histology (8758 cases; 46,126 controls) | | | Non-endometrioid histologies (1230 cases; 35,447 controls) | | | Between histologies |
|---|---|---|---|---|---|---|---|---|---|---|---|---|---|---|---|---|---|
| | | | | | | | Allelic OR (95% CI) | P | I² | BFDP (%) | Allelic OR (95% CI) | P | I² | Allelic OR (95% CI) | P | I² | P |
| 6p22.3 | rs1740828 | 21,648,854 | SOX4 | G:A | 0.52 | G | 1.15 (1.11, 1.19) | 4.2E−16 | 25% | <1% | 1.16 (1.11, 1.20) | 6.0E−13 | 11% | 1.00 (0.91, 1.10) | 9.81E−01E−01 | 7% | 0.016 |
| 6q22.31 | rs2747716 | 125,687,226 | HEY2, NCOA7 | A:G | 0.57 | 1.00 | 1.10 (1.07, 1.14) | 2.9E−10 | 55% | <1% | 1.12 (1.08, 1.16) | 4.4E−10 | 36% | 0.99 (0.91, 1.08) | 7.9E−01 | 0% | 0.058 |
| 8q24.21[b] | rs35286446 | 128,433,617 | MYC | GAT:G | 0.58 | 0.99 | 1.10 (1.06, 1.13) | 3.1E−09 | 0% | <1% | 1.10 (1.06, 1.14) | 1.8E−07 | 0% | 1.10 (1.01, 1.19) | 3.6E−02 | 0% | 0.83 |
| 8q24.21[b] | rs4733613 | 128,587,032 | MYC | C:G | 0.12 | G | 1.18 (1.13, 1.24) | 7.5E−14 | 0% | <1% | 1.21 (1.15, 1.28) | 1.2E−13 | 0% | 1.08 (0.95, 1.22) | 2.3E−01 | 0% | 0.041 |
| 8q24.21[b] | rs139584729 | 128,611,656 | MYC | C:G | 0.98 | 0.97 | 1.40 (1.25, 1.58) | 2.4E−08 | 0% | 2% | 1.48 (1.28, 1.70) | 7.6E−08 | 0% | 1.18 (0.86, 1.63) | 3.0E−01 | 0% | 0.24 |
| 13q22.1 | rs7981863 | 73,238,004 | KLF5, KLF12 | C:T | 0.72 | G | 1.16 (1.12, 1.20) | 2.7E−17 | 26% | <1% | 1.17 (1.13, 1.22) | 4.9E−15 | 0% | 1.13 (1.02, 1.24) | 1.4E−02 | 45% | 0.95 |
| *14q32.33[c]* | *rs2498796* | *104,776,883* | *AKT1* | *A:G* | *0.30* | *0.98* | *1.07 (1.03, 1.11)* | *6.3E−05* | *1%* | *98%* | *1.09 (1.04, 1.13)* | *3.2E−05* | *0%* | *1.07 (0.98, 1.17)* | *1.4E−01* | *11%* | *0.69* |
| 15q15.1 | rs937213 | 40,029,923 | EIF2AK4, BMF | C:T | 0.42 | G | 1.09 (1.06, 1.13) | 5.1E−09 | 0% | 1% | 1.12 (1.08, 1.16) | 6.9E−10 | 0% | 1.15 (1.06, 1.25) | 1.0E−03 | 12% | 0.78 |
| 15q21.2 | rs17601876 | 51,261,712 | CYP19A1 | G:A | 0.48 | 1.00 | 1.12 (1.09, 1.16) | 3.3E−14 | 0% | <1% | 1.12 (1.08, 1.16) | 2.3E−10 | 0% | 1.05 (0.96, 1.14) | 3.0E−01 | 35% | 0.02 |
| 17q12 | rs11263761 | 37,737,784 | HNF1B | A:G | 0.52 | 0.98 | 1.15 (1.12, 1.19) | 3.2E−20 | 25% | <1% | 1.15 (1.11, 1.19) | 3.4E−14 | 14% | 1.20 (1.10, 1.31) | 3.6E−05 | 2% | 0.70 |

EAF: effect allele frequency among control subjects in the UK Biobank, Info: imputation quality score for the OncoArray set, G: genotyped SNPs, I²: heterogeneity I² statistic, BFDP: Bayes false discovery[46]
[a]Position is with reference to build 38 of the reference genome
[b]Results for the 8q24 SNPs are from the conditional model containing all three SNPs. r² = 0.02 for rs35286446 and rs4733613; r² = 0.01 for rs35286446 and rs139584729; r² = 0.003 for rs4733613 and rs139584729
[c]rs2498796 (14q32.33) was not replicated in this analysis

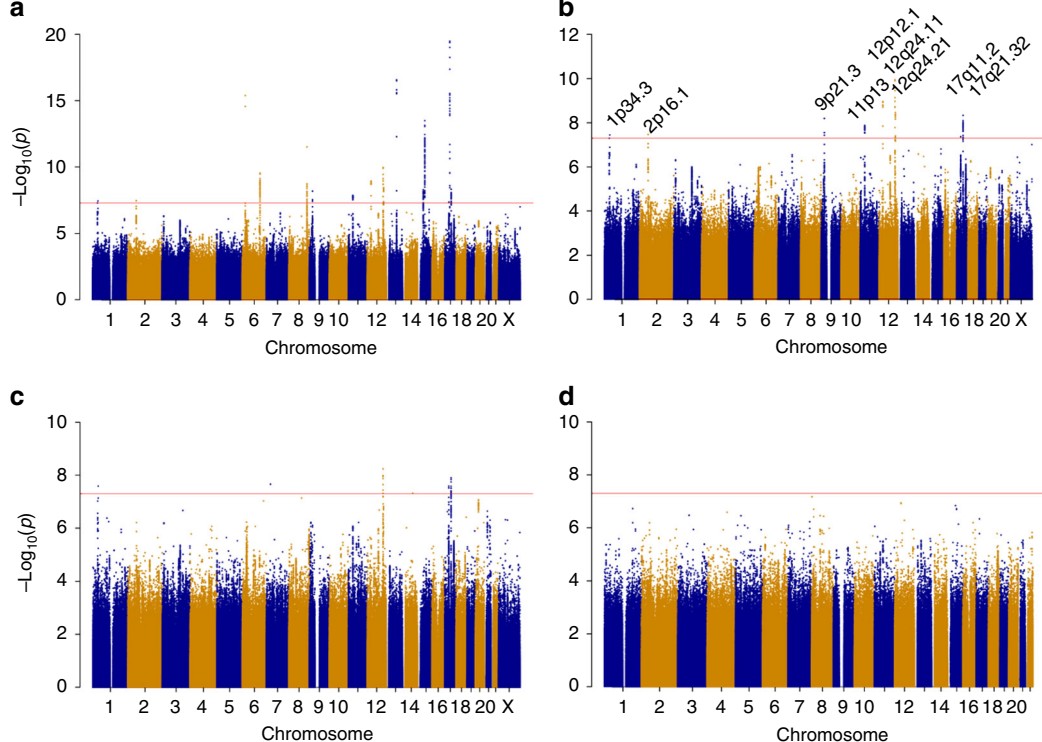

**Fig. 1** Manhattan plot of the results of the endometrial cancer meta-analysis of 12,906 cases and 108,979 controls. Genetic variants are plotted according to chromosome and position (x axis) and statistical significance (y axis). The red line marks the $5 \times 10^{-8}$ GWAS significance threshold. **a** Endometrial cancer (all histologies). **b** Endometrial cancer (all histologies) excluding variants within 500 kb of previously published endometrial cancer variants. **c** Endometrioid histology endometrial cancer excluding variants within 500 kb of previously published endometrial cancer variants. **d** Non-endometrioid histology endometrial cancer

There was no evidence of secondary signals at any of these nine loci after conditioning on the most significant variant. There was no significant between-study heterogeneity (minimum Cochran Q-test $P_{het} = 0.04$, maximum $I^2 = 41\%$, Supplementary Fig. 3), and random-effects meta-analyses produced very similar results (Supplementary Data 2). Twenty-five additional independent loci showed moderately significant ($P < 1 \times 10^{-6}$) associations, nine with endometrial cancer overall, nine specifically with endometrioid histology, and seven with non-endometrioid histology (Supplementary Data 2).

**Overlap with published GWAS associations**. Using a 100:1 likelihood ratio, "credible causal risk" variants (ccrSNPs) were compiled for each of the nine new endometrial cancer risk loci (Supplementary Data 3). These included 239 variants located in non-coding regions, 2 missense variants (rs2278868 SKAP1 Gly161Ser and rs3184504 SH2B3 Trp262Arg), and 1 synonymous variant (rs1129506 EVI2A Ser23Ser). Comparing to the NHGRI-EBI catalog of published GWAS, 37 SNPs previously associated with a cancer, hormonal trait, or anthropometric trait fall within 500 kb of any one of the novel endometrial cancer SNPs. However, the only overlap from the set of ccrSNPs with other traits was the colorectal and endometrial cancer susceptibility SNP rs3184504 in *SH2B3* (Supplementary Data 4).

**eQTL analyses**. LD score regression analyses using eQTL results from GTEx[12] showed that endometrial cancer heritability exhibited the strongest evidence for enrichment for variants associated with genes specifically expressed in vaginal and uterine tissue, in line with prior assumptions, although none of the tissue-specific enrichments were significant (weighted regression with jackknife standard errors) after Bonferroni correction, adjusting for the number of tissues tested (Supplementary Fig. 4). eQTL analyses were performed using data from a variety of tissue sources (Supplementary Methods), including endometrial tumor and adjacent normal endometrium tissue from The Cancer Genome Atlas (TCGA)[13], normal cycling endometrium[14] and, in view of the GTEx enrichment results, vaginal and uterine tissue. Additionally, we assessed eQTLs from whole blood[15], which provided substantially increased power over solid tissue analyses due to increased sample size. eQTLs were detected at five of the nine novel loci (Supplementary Data 3, Supplementary Data 5, Supplementary Figs. 5–13, Table 2).

**Gene network analysis**. Network analysis was performed using candidate causal genes identified in this study, in addition to candidate causal genes identified in previous studies[6–8] (Supplementary Data 6). One major network was identified, containing 18 of the 25 candidate causal genes (Supplementary Fig. 14). Network hubs included CCND1, CTNNB1, and P53, which are encoded by genes that are somatically mutated in endometrial cancer[13]. Analysis of the network revealed significant enrichment (Benjamini–Hochberg adjusted $P < 0.05$, hypergeometric test) in relevant pathways such as endometrial cancer signaling, adipogenesis, Wnt/β-catenin signaling, estrogen-mediated S-phase entry, P53 signaling, and PI3K/AKT signaling (Supplementary Data 7).

**Table 2 Meta-analysis results for newly identified genome-wide significant endometrial cancer risk loci**

| Region | SNP | Position (bp)[a] | Nearby or candidate gene(s)[b] | Effect: other alleles | EAF | Info | All histologies (12,906 cases; 108,979 controls) | | | | Endometrioid histology (8758 cases; 46,126 controls) | | | Non-endometrioid histologies (1230 cases; 35,447 controls) | | | Between histologies |
|---|---|---|---|---|---|---|---|---|---|---|---|---|---|---|---|---|---|
| | | | | | | | Allelic OR (95% CI) | $P$ | $I^2$ | BFDP (%) | Allelic OR (95% CI) | $P$ | $I^2$ | Allelic OR (95% CI) | $P$ | $I^2$ | $P$ |
| 1p34.3 | rs113998067 | 37,607,755 | *GNL2, RSPO1,* **CDCA8** | C:T | 0.04 | 0.90 | 1.23 (1.14, 1.32) | 3.6E−08 | 20% | 2% | 1.27 (1.17, 1.38) | 2.6E−08 | 33% | 1.21 (0.98, 1.50) | 7.0E−02 | 0% | 0.99 |
| 2p16.1 | rs148261157 | 60,670,444 | *BCL11A* | A:G | 0.03 | 0.88 | 1.26 (1.16, 1.36) | 3.4E−08 | 16% | 2% | 1.25 (1.14, 1.38) | 4.7E−06 | 21% | 1.64 (1.32, 2.04) | 9.6E−06 | 0% | 0.0026 |
| 9p21.3 | rs1679014 | 22,207,038 | *CDKN2A, CDKN2B* | T:C | 0.07 | G | 1.18 (1.12, 1.25) | 6.4E−09 | 0% | <1% | 1.17 (1.09, 1.25) | 4.4E−06 | 0% | 1.19 (1.02, 1.38) | 3.0E−02 | 6% | 0.14 |
| 11p13 | rs10835920 | 32,468,118 | *WT1,* **WT1-AS,** *RCN1,* *CCDC73, EIF3M, TCP11L1* | T:C | 0.38 | 0.99 | 1.09 (1.06, 1.13) | 1.3E−08 | 0% | 1% | 1.10 (1.05, 1.14) | 2.1E−06 | 0% | 1.10 (1.01, 1.20) | 3.8E−02 | 15% | 0.68 |
| 12p12.1 | rs9668337 | 26,273,405 | *SSPN* | A:G | 0.74 | 0.99 | 1.11 (1.08, 1.15) | 1.1E−09 | 0% | <1% | 1.10 (1.06, 1.15) | 2.6E−06 | 0% | 1.10 (1.00, 1.22) | 5.1E−02 | 0% | 0.88 |
| 12q24.11 | rs3184504 | 111,446,804 | **SH2B3** | C:T | 0.52 | G | 1.10 (1.07, 1.14) | 1.1E−10 | 0% | <1% | 1.11 (1.07, 1.15) | 5.8E−09 | 0% | 1.10 (1.01, 1.19) | 3.2E−02 | 4% | 0.79 |
| 12q24.21 | rs10850382 | 114,776,743 | *LOC107984437* | T:C | 0.31 | G | 1.10 (1.07, 1.14) | 3.5E−09 | 0% | <1% | 1.11 (1.07, 1.15) | 1.5E−07 | 0% | 1.02 (0.93, 1.12) | 6.7E−01 | 0% | 0.16 |
| 17q11.2 | rs1129506 | 31,319,014 | **EVI2A, NF1, SUZ12,** *RP11-848P1.5* | G:A | 0.38 | G | 1.10 (1.06, 1.13) | 4.3E−08 | 0% | 4% | 1.11 (1.07, 1.15) | 1.3E−07 | 36% | 1.07 (0.98, 1.17) | 1.3E−01 | 13% | 0.27 |
| 17q21.32 | rs882380 | 48,216,874 | *SKAP1,* **SNX11** | A:C | 0.61 | 0.99 | 1.10 (1.06, 1.13) | 4.7E−09 | 41% | <1% | 1.11 (1.07, 1.15) | 1.2E−08 | 34% | 1.08 (0.99, 1.18) | 7.6E−02 | 0% | 0.62 |

EAF: effect allele frequency among control subjects in the UK Biobank, Info: imputation quality score for the OncoArray set, G: genotyped SNPs, $I^2$: heterogeneity $I^2$ statistic, BFDP: Bayes false discovery probability[46]
[a]Position is with reference to build 38 of the reference genome
[b]Bolded genes are candidate genes identified from eQTL analysis

**Functional annotation of ccrSNPs.** Next, ccrSNPs were mapped to epigenomic features from endometrial cancer cell lines (Supplementary Data 3, Supplementary Figs. 5–13). Chromatin immunoprecipitation (ChIP-seq) was used to map histone modifications indicative of promoters or enhancers (H3K4Me1, H3K4Me3, and H3K27Ac) in two endometrial cancer cell lines (Ishikawa and JHUEM-14). Mapping of DNaseI hypersensitivity sites (indicative of open chromatin) and ChIP-seq data for transcription factor binding sites from Ishikawa cells were accessed from ENCODE[16]. We also included mapping of H3K427Ac histone modifications for uterus and vagina from ENCODE. Overall, 73% of ccrSNPs overlapped at least one epigenomic feature, including at least one ccrSNP per novel risk region. This overlap was significantly greater than the overlap observed for these epigenomic features with ccrSNPs related to, for example, endometriosis[17] (51%; Fisher's exact $P = 8.7 \times 10^{-8}$) or schizophrenia[18] (40%; Fisher's exact $P < 2.2 \times 10^{-16}$). These findings indicate the relevance of the selected cell and tissue types for informing endometrial cancer biology and a role for the assessed epigenomic features in regulatory processes related to the ccrSNPs. Overlaps between ccrSNPs and epigenomic features increased significantly after stimulation with estrogen (50% versus 38% for unstimulated features; Fisher's exact $P = 5.6 \times 10^{-3}$), emphasizing the importance of estrogen in endometrial cancer etiology.

**Mendelian randomization analyses.** This expanded meta-analysis allowed us to strengthen our previous Mendelian randomization findings[19,20] that higher body mass index (BMI) ($P = 1.7 \times 10^{-11}$, two-sample inverse-variance weighted Mendelian randomization (MR) test), but not waist:hip ratio ($P = 0.71$), is causal for endometrial cancer (Table 3) and that the protective effect of later menarche on endometrial cancer risk (OR = 0.82, 95% CI 0.77–0.87 per year of delayed menarche, $P = 2.2 \times 10^{-9}$) is partially mediated by the known relationship between lower BMI and later menarche, with a more modest protective effect of

later menarche after adjusting for genetically predicted BMI (OR = 0.88, 95% CI 0.82–0.94, $P = 3.8 \times 10^{-4}$). The association between genetically predicted age at natural menopause and endometrial cancer did not reach statistical significance (OR = 1.03, 95% CI 1.00–1.06, $P = 0.060$). In contrast to the reported effects for breast and prostate cancer[21,22], we found no evidence that genetically predicted adult height is associated with endometrial cancer ($P = 0.90$).

**Genetic correlation analyses.** Cross-trait LD score regression of 224 non-cancer traits available via the LD Hub interface[23], identified significant genetic correlations between endometrial cancer and 14 traits. All of these are either a measure of obesity or are strongly and significantly (correlation-corrected jackknife $P < 10^{-12}$) genetically correlated with BMI (i.e., age of menarche, type 2 diabetes, and years of schooling) (Supplementary Data 8), in line with the established relationship between obesity and endometrial cancer risk.

**Discussion**

In the largest GWAS meta-analysis assessing endometrial cancer risk, we discovered nine new genetic risk regions. We also confirmed the association of genetic variants with endometrial cancer risk at seven of the eight previously published genetic risk regions for this disease[5–8]. Using this larger GWAS-meta dataset, we were also able to confirm the previously published Mendelian randomization studies finding that higher BMI is causal for endometrial cancer risk[20], and the protective effect of later age of menarche on endometrial cancer risk[19]. Genetic correlation analyses also indicated a relationship between endometrial cancer and obesity-related traits.

Candidate causal genes identified through eQTLs included *CDCA8* (1p34.3), a putative ovarian cancer oncogene[24], which encodes an essential regulator of mitosis and cell division[25]; *RCN1* (11p13), encoding a calcium-binding protein that binds onco-proteins such as JAK2[26] and MYC[27]; *WT1-AS* (11p13), a long

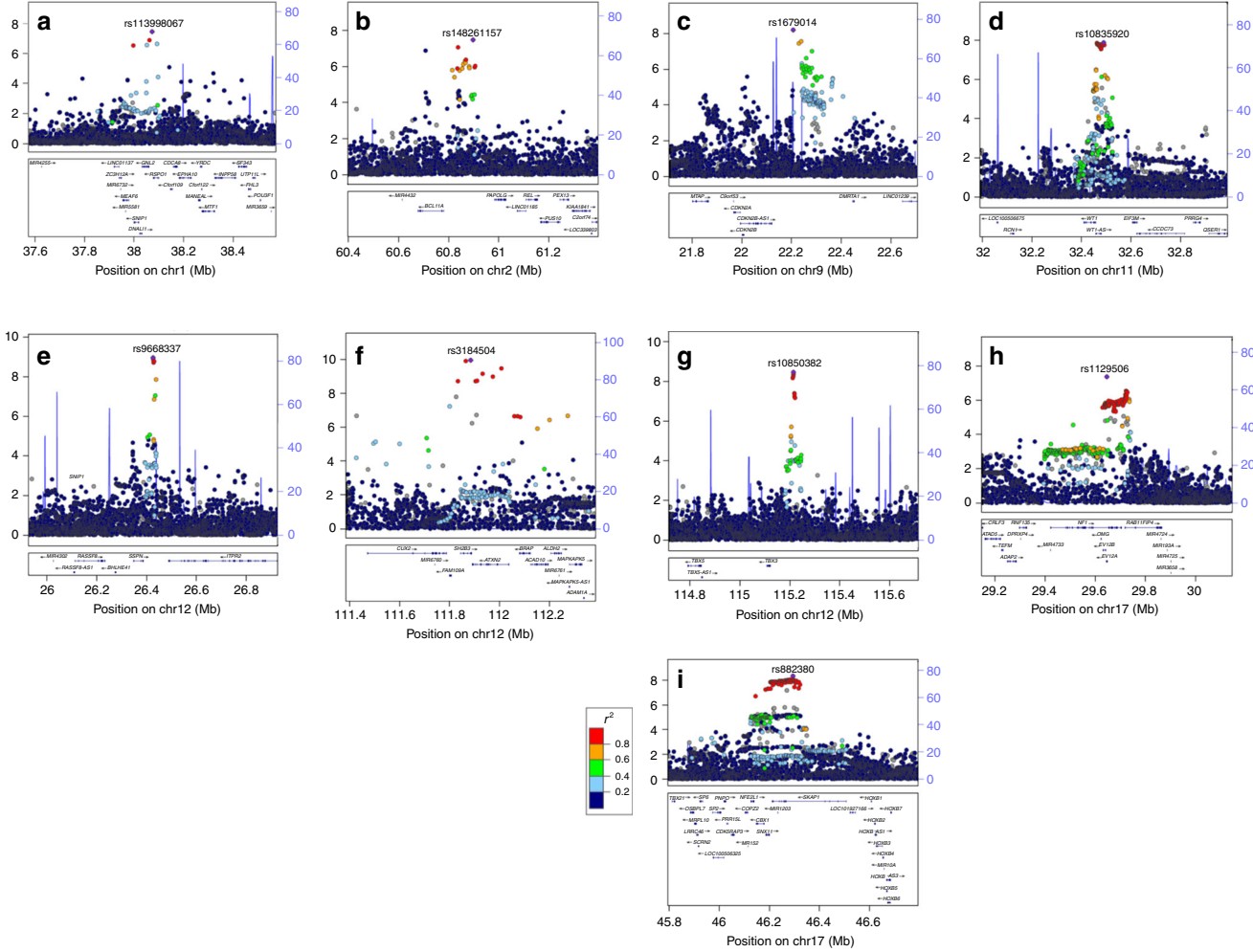

**Fig. 2** Regional association plots for the nine novel endometrial cancer loci. $-\log_{10}(p)$ from the fixed-effects meta-analysis is on the left $y$ axis, recombination rate (cM/Mb) is on the right $y$ axis (plotted as blue lines). The color of the circles shows the level of linkage disequilibrium between each variant and the most significantly associated variant (purple diamond) ($r^2$ from the 1000 Genomes 2014 EUR reference panel—see key). Genes in the region are shown beneath each plot. **a** 1p34.3, **b** 2p16.1, **c** 9p21.3, **d** 11p13, **e** 12p12.1, **f** 12q24.11, **g** 12q24.21, **h** 17q11.2, **i** 17q21.32

non-coding RNA that regulates the *WT1* oncogene[28,29]; *SH2B3* (12p24.11) encoding a negative regulator of the oncogenic KIT and JAK2 signal transduction proteins[30]; and tumor suppressor gene *NF1* (17q11.2) encoding a negative regulator of RAS-mediated signal transduction[31], which acquires putative driver mutations in TCGA endometrial tumors (http://www.cbioportal.org/study?id=ucec_tcga). Notably, the highly significant eQTL associations between ccrSNPs and expression of *SH2B3* (linear regression $P \geq 5.62 \times 10^{-20}$) and *NF1* ($P \geq 1.32 \times 10^{-56}$) in blood revealed risk alleles to be associated with decreased gene expression for both loci, consistent with the role of these genes in tumor development.

Intersections of ccrSNPs with epigenomic marks mapped in endometrial cancer cell lines, uterine tissue, and vaginal tissue found more endometrial cancer ccrSNPs overlapped with these features than ccrSNPs for endometriosis[17] or schizophrenia[18]. These findings highlight the relevance of these tissues for functional studies of endometrial cancer biology. Given the established role of estrogen in endometrial carcinogenesis[32], it is perhaps not surprising that endometrial cancer ccrSNPs exhibited greater overlap with epigenomic features present after estrogen stimulation. However, this finding provides evidence that

functional studies of endometrial cancer should be performed under these conditions.

Using LD score regression, we estimated that ~28% of the approximately twofold FRR of endometrial cancer could be explained by variants, which can be reliably imputed from OncoArray genotypes. The common endometrial cancer variants identified to date together explain up to 6.8% of the FRR, including 2.7% contributed by the nine additional variants reported here; this may be an overestimate, given that the ORs for the new loci likely include some upwards bias (the so-called winner's curse). In summary, we have doubled the number of endometrial cancer risk loci, explaining around one quarter (6.9%/28%) of the portion of the FRR attributable to common, readily-imputable SNPs. Furthermore, eQTL analyses have identified candidate causal genes and pathways related to tumor development for follow-up studies that will provide further insight into endometrial cancer biology.

## Methods
**Study samples.** Analyses were based on 13 studies of endometrial cancer, of which four studies contributed case samples to more than one genotyping project. Data were also included from the E2C2 consortium of 45 separate studies. All

**Table 3 Effects of genetically predicted anthropometric and reproductive traits on risk of endometrial cancer**

| Trait | Instrumental variable SNPs | Endometrial cancer All histology (OR and 95% CI) | Endometrial cancer Endometrioid histology (OR and 95% CI) | Endometrial cancer Non-endometrioid histology (OR and 95% CI) |
|---|---|---|---|---|
| Body mass index (BMI)[a] | 77 SNPs[49] | 1.92 (1.63, 2.25), $P = 1.7E-11$ | 2.04 (1.69, 2.46), $P = 8.6E-11$ | 1.65 (1.13, 2.41), $P = 0.011$ |
| Waist:hip ratio | 47 SNPs[50] | 0.95 (0.72, 1.25), $P = 0.71$ | 0.94 (0.71, 1.24), $P = 0.66$ | 1.27 (0.69, 2.33), $P = 0.45$ |
| Age at menarche (years); total effect | 368 SNPs[19] | 0.82 (0.77, 0.87), $P = 2.2E-9$ | 0.80 (0.74, 0.86), $P = 1.9E-9$ | 0.93 (0.79, 1.08), $P = 0.33$ |
| Age at menarche (years); direct effect[b] | 368 SNPs[19], BMI weights from Locke et al.[49] | 0.88 (0.82, 0.94), $P = 3.8E-4$ | 0.86 (0.79, 0.93), $P = 2.7E-4$ | 0.97 (0.82, 1.16), $P = 0.76$ |
| Age at natural menopause (years) | 54 SNPs[53] | 1.03 (1.00, 1.06), $P = 0.060$ | 1.02 (0.99, 1.06), $P = 0.19$ | 1.07 (0.99, 1.14), $P = 0.075$ |
| Adult height | 814 SNPs[51, 52] | 1.00 (0.95, 1.06), $P = 0.90$ | 0.99 (0.93, 1.05), $P = 0.63$ | 1.00 (0.88, 1.15), $P = 0.95$ |

Odds ratios (ORs) are per year for age at menarche and for age at natural menopause, but are not in meaningful units for the other traits because the published SNP-trait associations are in terms of inverse-rank normalized residuals
[a]Note, none of the endometrial cancer risk variants identified to date have been directly related to BMI-associated SNPs, or BMI monogenic disorders
[b]The direct effect of age at menarche on endometrial cancer risk is adjusted for the mediating effect of genetically predicted BMI[54]

participants were of European ancestry. Data from the E2C2 genome-wide association studies (GWAS) and from the ANECS, SEARCH, NSECG GWASs and the iCOGS project have been previously published, and are described in de Vivo et al.[33] and Cheng et al.[6], respectively.

**The OncoArray study**. The "OncoArray" genotyping chip[34] contains 533,631 variants, around half of which were selected to provide a "GWAS backbone," with the remaining variants selected on the basis of prior evidence of association with cancer or a cancer-related trait. The OncoArray chip was used to genotype 5061 endometrial cancer cases from ten studies in Australia, Belgium, Germany, Sweden, UK, and USA. Genotyping was carried out at two sites: the Center for Inherited Disease Research (CIDR; nine studies) and The University of Melbourne (one study). Details of the genotype calling are given in Amos et al.[34]

SNP-wise QC was conducted using genotype data from all consortia participating in the OncoArray experiment[34]. SNPs with call rate <95% in any of the consortia, SNPs not in Hardy–Weinberg equilibrium (HWE) ($P < 10^{-7}$ in controls and $P < 10^{-12}$ in cases) and SNPs with concordance <98% among 5280 duplicate pairs of samples were excluded, leaving 483,972 SNPs. Prior to imputation, SNPs with minor allele frequency (MAF) <1% and call rate <98% in any consortium were also excluded, as were SNPs that could not be linked to the 1000 Genomes Project reference panel or for which the MAF differed significantly from the European reference panel frequency. A further 1128 SNPs were excluded after review of cluster plots, hence 469,364 SNPs were used in the imputation.

The 5061 OncoArray-genotyped endometrial cancer cases were country-matched to controls who had been genotyped in an identical process as part of the Breast Cancer Association Consortium[35,36]. Samples with call rate <95%, with excessively low or high heterozygosity or with an estimated proportion of European ancestry <80% (based on a principal components analysis of 2318 informative markers and with reference to the HapMap populations) were excluded, as were suspected males and individuals who were XO or XXY.

Duplicates and close relatives were identified from estimated genomic kinship matrices. Pairwise comparisons were made among all samples genotyped as part of the OncoArray, iCOGS, or ANECS/SEARCH/NSECG GWAS genotyping projects. Where pairs of duplicates or close relatives were identified between projects, the sample with the more recent genotyping was retained, hence the numbers of cases included here from the ANECS/SEARCH/NSECG GWASs and iCOGS projects are lower than in the original publications. For case–control pairs from within the same project, the case was preferentially retained, and for case–case or control–control pairs, the sample with the higher call rate was used. Following these exclusions, OncoArray genotypes from 4710 cases and 19,438 controls were included in the analyses.

All OncoArray samples (along with all samples from the ANECS/SEARCH/ NSECG GWASs and the iCOGS project) were imputed using the October 2014 (version 3) release of the 1000 Genomes Project reference panel. Samples were phased using SHAPEITv2[37] and genotypes were imputed using the IMPUTEv2[38] software for non-overlapping 5-Mb intervals. Analyses were restricted to the ~11.4 million SNPs with MAF >0.5% and $r^2 > 0.4$.

**Other studies**. The 2695 cases and 2777 controls from the E2C2 consortium were genotyped using the Illumina Human OmniExpress array (2271 cases, 2219 controls from the United States) or the Illumina Human 660W array (424 cases, 558

controls from Poland)[33] and both sets were separately imputed to the 1000 Genomes Project v3 reference panel using "minimac2" software, following standard quality control steps[38,39].

The 288 cases from six population-based case–control studies within the Women's Health Initiative were genotyped using five different arrays (Supplementary Data 1) and were each separately imputed using the combined 1000 Genome Project v3 and UK10K reference panels using "minimac2" software[39], following standard quality measures and the exclusion of SNPs with a MAF <1%. Five controls for each case were selected randomly, matched on study.

Data were also included from the first phase of UK Biobank genotyping, comprising 636 Cancer Registry-confirmed endometrial cancer cases (as of October 2016) and 62,853 cancer-free female controls. Samples were genotyped using Affymetrix UK BiLEVE Axiom array and Affymetrix UK Biobank Axiom® array and imputed to the combined 1000 Genome Project v3 and UK10K reference panels using SHAPEIT3[40] and IMPUTE3[41].

No analyses to identify duplicates or relatives between samples from the E2C2, WHI, or UK Biobank studies, and any other study were carried out. However, given the sampling frame of these studies, it is very unlikely that there would have been any meaningful sample overlap.

After QC exclusions, the analysis included 12,906 endometrial cancer cases (3613 of which have not been included in any previous publication) and 108,979 controls. Analyses were also carried out specifically for endometrial cancer of endometrioid histology (8758 cases) and endometrial cancer with non-endometrioid histology (1230 cases). Exploratory analyses for specific non-endometrioid histologies (serous carcinoma, carcinosarcoma, clear cell carcinoma, and mucinous carcinoma) included a small number of cases of mixed histotype, where the major component was non-endometrioid. The UK Biobank data did not include information about histology.

All participating studies were approved by research ethics committees from QIMR Berghofer Medical Research Institute, University-Clinic Erlangen, Karolinska Institutet, UZ Leuven, The Mayo Clinic, The Hunter New England Health District, The Regional Committees for Medical and Health Research Ethics Norway, and the UK National Research Ethics Service (04/Q0803/148 and 05/ MRE05/1). All participants provided written, informed consent.

**Statistical analyses**. Per-allele ORs and the s.e. of the logORs were computed using logistic regression for each of the ANECS, SEARCH, NSECG, WHI, and UK Biobank GWASs, for the two E2C2 GWASs and, by country, for the iCOGS and OncoArray studies, giving a total of 17 strata. Case-only analyses were used to assess heterogeneity in SNP effects by histology (endometrioid histology versus non-endometrioid histology). In the OncoArray analysis, potential population stratification was adjusted for using the first nine principal components; these were estimated using data for 33,661 uncorrelated SNPs with MAF >0.05 and pairwise $r^2 < 0.1$ (including 2318 SNPs specifically selected as informative for continental ancestry) using purpose-written software (http://ccge.medschl.cam.ac.uk/software/ pccalc). Other studies were similarly adjusted for their relevant principal components.

Analyses were carried out using SNPTEST[42] for the ANECS, SEARCH, and NSECG GWASs, using ProbABEL[43] for the E2C2 GWASs, and using in house software for the iCOGS, OncoArray, WHI, and UK Biobank studies. We assessed residual population stratification by computing the test statistic inflation

adjusted to a sample size of 1000 cases and 1000 controls ($\lambda_{1000}$'s), both overall and with each strata, using 33,278 uncorrelated SNPs ($r^2 < 0.1$). The overall $\lambda_{1000}$ was 1.004, with stratum-specific $\lambda_{1000}$'s between 0.996 and 1.128 (observed for the smallest strata, the German iCOGS dataset; Supplementary fig. 1).

The estimated ORs from the different studies were combined in a fixed-effects inverse-variance weighted meta-analysis using the "meta" software[44]. For each variant, results from any strata for which the imputation information score was <0.4, the MAF <0.005 or the OR >3 or <0.333 were excluded. Following the meta-analysis, SNPs with valid results in fewer than two of the strata, or with between-strata heterogeneity $P < 5 \times 10^{-8}$ were also excluded, leaving 11.7 million SNPs. A random-effects meta-analysis was also carried out.

Using the conventional $5 \times 10^{-8}$ genome-wide significance threshold, all SNPs lying within ± 500 kb of a significant SNP were initially considered as part of that locus. Approximate conditional analysis in the GCTA program[11,45] with an LD reference panel of 4000 OncoArray-genotyped control subjects were then used to look for additional independently-associated SNPs within each locus. Only uncorrelated ($r^2 < 0.05$) secondary signals were included. The only locus with evidence of significant signals after conditioning on the most strongly associated SNP was the previously published 8q24 locus[6] (Table 1). For each locus, the set of credible causal risk SNPs (ccrSNPs) was defined as those variants within ± 500 kb of the most significant SNP and for which the likelihood from the association analysis was no less than one hundredth the likelihood of the most significant SNP (i.e., odds of <1 : 100). A BFDP for each significant SNP was estimated on the basis of a maximum plausible OR of 1.5 and a prior probability of association of 0.0001[46].

The proportion of the FRR of endometrial cancer due to the identified variants was estimated using a log-additive model, where $p_j$, $\beta_j$, and $\tau_j$ are the MAF, logOR, and se(logOR), respectively for variant $j$, and $\lambda = 2$ is the reported FRR of endometrial cancer. The effect estimates used were those estimated in the current study, both for the new loci and for the loci replicated from previous studies.

$$\text{Proportion FRR} = \frac{1}{\ln(\lambda)} \sum_j p_j (1 - p_j)(\beta_j^2 - \tau_j^2).$$

The proportion of the endometrial cancer FRR that can be explained by all SNPs is given by the frailty-scale heritability, $h_f^2$, divided by $2\ln(\lambda)$. This was estimated using LD score regression[47], based on the full set of meta-analysis summary estimates, restricted to those SNPs present on the HapMap v3 dataset with MAF >1% and imputation quality $R^2 > 0.9$ in the OncoArray imputation using the 1000 Genomes Phase 3 reference panel. The frailty-scale heritability (as opposed to the observed-scale heritability) was obtained by replacing the total sample, $N$, for each study with an effective sample size $N_j$ for SNP $j$, which effectively weights each SNP according to its frequency and the variance of the effect estimate, i.e.,

$$N_j = \frac{1}{2p_j\left(1 - p_j\right)\tau_j^2}.$$

Cross-trait LD score regression via the LD Hub interface (28 September 2017, v1.4.1) was used to estimate the genetic correlations between endometrial cancer and 224 traits from 24 categories[23].

The casual effects of five anthropometric or reproductive factors on the risks of endometrial cancer were estimated using two-sample summary statistic inverse-variance weighted MR analyses[48]. Instrumental variables for each factor consisted of the most recent set of published GWAS-significant SNPs for that trait; 77 SNPs for body mass index (BMI)[49], 47 SNPs for waist:hip ratio[50], 814 SNPs for adult height[51,52], 54 SNPs for age at natural menopause[53], and 368 SNPs for age at menarche[19]. A multivariable MR adjusting for the effects of the 368 menarche SNPs on BMI (a potential mediator) was used to estimate the direct effect of menarche on endometrial cancer, not via BMI[54].

**Cell culture.** Ishikawa and JHUEM-14 cells were a gift from Prof PM Pollock (Queensland University of Technology). Cell lines were authenticated using STR profiling and confirmed to be negative for mycoplasma contamination. Ishikawa cells were cultured in Dulbecco's modified Eagle's medium (DMEM; Life Technologies #1195-065) with 10% fetal bovine serum (FBS) and antibiotics (100 IU/ml penicillin and 100 µg/ml streptomycin). JHUEM-14 cells were cultured in DMEM/F12 medium (Life Technologies #11320-033) with 10% FBS and antibiotics.

**Cell fixing and chromatin shearing.** Ishikawa and JHUEM-14 cells were plated on to 10-cm tissue culture dishes in phenol red-free DMEM (Sigma-Aldrich #D1145) supplemented with L-glutamine, sodium pyruvate, and 10% charcoal-dextran-stripped FBS. Three days later, media were replaced and cells incubated with fresh medium containing either 10 nM estradiol or DMSO (vehicle control) for 3 h. Cells were washed twice with PBS and fixed at room temperature in 1% formaldehyde in PBS. After 10 min, cells were placed on ice and washed twice with ice-cold PBS. The reaction was quenched with 10 mM DTT in 100 mM Tris-HCl (pH 9.4) and cells removed from the dish with a cell scraper. Cells were incubated at 30 °C for 15 min, then spun for 5 min at 2000×g. Cells were washed

sequentially with ice-cold PBS, buffer I (0.25% Triton X-100, 10 mM EDTA, 0.5 mM EGTA, 10 mM HEPES, pH 6.5) and buffer II (200 mM NaCl, 1 mM EDTA, 0.5 mM EGTA, 10 mM HEPES, pH 6.5) and centrifuged for 5 min at 2000×g at 4 °C. Cells were resuspended in 300–750 µl of lysis buffer (1% SDS, 10 mM EDTA, 50 mM Tris-HCl, pH 8.1, with complete protease inhibitor cocktail (Sigma-Aldrich #11836145001)). Ishikawa cells were sonicated for eight cycles (10 s) and JHUEM-14 cells for 20 cycles using the highest power setting of a Branson Digital Sonifier SLPt. After chromatin shearing was confirmed by agarose gel electrophoresis, samples were centrifuged for 10 min at 4 °C.

**Chromatin immunoprecipitation and sequencing.** Samples were diluted 10-fold in 1% Triton X-100, 2 mM EDTA, 20 mM Tris.HCl (pH 8.1), and 150 mM NaCl with complete protease inhibitor cocktail. Magna ChIP protein A/G magnetic beads (EMD Millipore #16-663) were added to 500 µl of diluted chromatin and incubated with 5 µg of antibody overnight at 4 °C. Antibodies to H3K4Me1 (Abcam #ab8895), H3K4Me3 (Abcam #ab8580), and H3K27Ac (Abcam #ab4729) were used (Supplementary Table 1). The next day supernatant was removed and the beads washed three times with the following ice-cold buffers: RIPA 150 (0.1% SDS, 1% Triton X-100, 1 mM EDTA, 50 mM Tris-HCl (pH 8.10, 150 mM NaC1, 0.1% sodium deoxycholate), RIPA 500 (0.1% SDS, 1% Triton X-100, 1 mM EDTA, 50 mM Tris-HCl (pH 8.10, 500 mM NaCl, 0.1% sodium deoxycholate), LiCl RIPA (500 mM LiCl, 1% NP-40, 1% deoxycholate, 1 mM EDTA, 50 mM Tris-HCl (pH 8.1)), and TE buffer. Chromatin was then eluted by incubating beads overnight at 60 °C with 100 µl of elution buffer (1% SDS, 100 mM NaHCO₃) and 0.5 mg/ml proteinase K. The next day beads were incubated at 95 °C for 10 min and supernatant removed. Chromatin was purified using the QIAquick Spin kit (QIAGEN) and eluted from columns using 50 µl of QIAGEN EB buffer. DNA was quantified using a Qubit dsDNA HS Assay kit (ThermoFisher Scientific).

Samples from two biological replicates for each treatment were sent to the Australian Genome Research Facility (Melbourne, Australia) for library preparation and sequencing (Illumina HiSeq 2500) with 50 bp reads. Mapping and analysis of ChIP-seq reads were performed using the ENCODE analysis pipeline, histone ChIP-seq Unary Control (GRCh37), with DNAnexus software tools (https://dnanexus.com). Replicated peaks across biological replicates were used for downstream analyses.

**eQTL analyses.** Summary eQTL results for non-cancer tissue were obtained using uterine ($N = 70$) and vaginal ($N = 79$) tissue-specific data generated by the Genotype-Tissue Expression Project (GTEx)[12], an endometrium eQTL dataset ($N = 123$) provided by Fung et al.[14], and a blood eQTL dataset (males and females; $N = 5311$)[15].

Data from endometrial cancer tumors and adjacent normal endometrial tissue were accessed from The Cancer Genome Atlas[13]. Patient germ line SNP genotypes (Affymetrix 6.0 arrays) and tissue expression RNA-seq data were downloaded through the controlled access portal, while epidemiological and tumor tissue copy-number data were downloaded through the public access portal. RNA-seq data were aligned and expression quantified to reads per kilobase per million (RPKM) as described in Painter et al.[10] and quality control performed on the germ line SNP genotypes as per Carvajal-Carmona et al.[55] Complete genotype, RNA-seq, and copy-number data were available for 277 genetically European patients (218 with endometrioid histology, 29 with adjacent normal tissue).

Germ line genotypes underwent further quality control before imputation to the 1000 Genomes Phase 3v5 reference panel by Eagle v2.3[56], using the Michigan Imputation Server[57]. Briefly, subjects were removed for genotype missingness >10% and SNPs were removed for missingness >10%, MAF <5%, and HWE P-value <5 × 10⁻⁸. SNPs were also removed if they were indels or non-biallelic variants, were ambiguous SNPs with a MAF >40%, were not matched to the reference panel, had a MAF difference with the reference panel of >20%, or were duplicates.

Genes with a median expression level of 0 RPKM across samples were removed, and the RPKM values of each gene were log2-transformed and samples were quantile normalized. The expression of the genes located within a 2-Mb window surrounding the ccrSNP at each of the newly identified risk loci were extracted from the expression dataset.

The associations between ccrSNPs and gene expression in all endometrial cancer tumor tissues, endometrioid endometrial cancer tissues only, and adjacent normal endometrial tissue, were evaluated using linear regression models using the MatrixEQTL program in R[58], adjusting for sequencing platform. Tumor tissue expression was also adjusted for copy-number variation, as previously described in Li et al.[59] A false discovery rate of <20% was used to report eQTL results from all datasets, except for the endometrium eQTL dataset where we used a P-value <0.01.

**Candidate causal gene network analysis.** Candidate causal genes identified in our previous studies and from the eQTL results in the current study (Supplementary Table 6) were analyzed using Ingenuity Pathway Analysis (QIAGEN;

accessed on 23 March 2018 and available at www.qiagen.com/ingenuity) to define gene networks and enrichment of genes in canonical signaling pathways.

**Data availability**. OncoArray germ line genotype data for the ECAC studies and E2C2 germ line genotype data have been deposited through the database of Genotypes and Phenotypes (dbGaP; accession number phs000893.v1.p1). Meta-GWAS summary statistics are available from the authors by request. Genotype data for non-cancer controls were provided by the Breast Cancer Association Consortium (BCAC) by application to the BCAC Data Access Coordination Committee (http://bcac.ccge.medschl.cam.ac.uk/). ChIP-seq data are available from the Gene Expression Omnibus (GEO; http://www.ncbi.nlm.nih.gov/geo/) under accession number GSE113818.

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

## Acknowledgements

We thank the many individuals who participated in this study and the numerous institutions and their staff who supported recruitment, detailed in full in the Supplementary Information. The iCOGS and OncoArray endometrial cancer analysis were supported by NHMRC project grants (ID#1031333 and ID#1109286) to A.B.S., D.F.E., A.M.D., D.J.T., and I.T. A.B.S. (APP1061779), P.M.W., and T.A.O'.M. (APP1111246) are supported by the NHMRC Fellowship scheme. A.M.D. was supported by the Joseph Mitchell Trust. I.T. is supported by Cancer Research UK and the Oxford Comprehensive Biomedical Research Center. Funding for the iCOGS infrastructure came from: the European Community's Seventh Framework Programme under grant agreement no. 223175 (HEALTH-F2-2009-223175) (COGS), Cancer Research UK (C1287/A10118, C1287/A10710, C12292/A11174, C1281/A12014, C5047/A8384, C5047/A15007, C5047/A10692, C8197/A16565), the National Institutes of Health (CA128978) and Post-Cancer GWAS initiative (1U19 CA148537, 1U19 CA148065 and 1U19 CA148112—the GAME-ON initiative), the Department of Defence (W81XWH-10-1-0341), the Canadian Institutes of Health Research (CIHR) for the CIHR Team in Familial Risks of Breast Cancer, Komen Foundation for the Cure, the Breast Cancer Research Foundation, and the Ovarian Cancer Research Fund. OncoArray genotyping of ECAC cases was performed with the generous assistance of the Ovarian Cancer Association Consortium (OCAC). We particularly thank the efforts of Cathy Phelan. The OCAC OncoArray genotyping project was funded through grants from the US National Institutes of Health (CA1X01HG007491-01 (Christopher I. Amos), U19-CA148112 (Thomas A. Sellers), R01-CA149429 (Catherine M. Phelan) and R01-CA058598 (Marc T. Goodman)); Canadian Institutes of Health Research (MOP-86727 (Linda E. Kelemen)); and the Ovarian Cancer Research Fund (Andrew Berchuck). CIDR genotyping for the Oncoarray was conducted under contract 268201200008I. OncoArray genotyping of the BCAC controls was funded by Genome Canada Grant GPH-129344, NIH Grant U19 CA148065, and Cancer UK Grant C1287/A16563.

ANECS recruitment was supported by project grants from the NHMRC (ID#339435), The Cancer Council Queensland (ID#4196615), and Cancer Council Tasmania (ID#403031 and ID#457636). SEARCH recruitment was funded by a programme grant from Cancer Research UK (C490/A10124). Stage 1 and stage 2 case genotyping was supported by the NHMRC (ID#552402, ID#1031333). Control data were generated by the Wellcome Trust Case Control Consortium (WTCCC), and a full list of the investigators who contributed to the generation of the data is available from the WTCCC website. We acknowledge use of DNA from the British 1958 Birth Cohort collection, funded by the Medical Research Council grant G0000934 and the Wellcome Trust grant 068545/Z/02—funding for this project was provided by the Wellcome Trust under award 085475. NSECG was supported by the EU FP7 CHIBCHA grant, Wellcome Trust Centre for Human Genetics Core Grant 090532/Z/09Z, and CORGI was funded by Cancer Research UK. We thank Nick Martin, Dale Nyholt, and Anjali Henders for access to GWAS data from QIMR Controls. Recruitment of the QIMR controls was supported by the NHMRC. The University of Newcastle, the Gladys M Brawn Senior Research Fellowship scheme, The Vincent Fairfax Family Foundation, the Hunter Medical Research Institute, and the Hunter Area Pathology Service all contributed toward the costs of establishing the Hunter Community Study. The Bavarian Endometrial Cancer Study (BECS) was partly funded by the ELAN fund of the University of Erlangen. The Hannover-Jena Endometrial Cancer Study was partly supported by the Rudolf Bartling Foundation. The Leuven Endometrium Study (LES) was supported by the Verelst Foundation for endometrial cancer. The Mayo Endometrial Cancer Study (MECS) and Mayo controls (MAY) were supported by grants from the National Cancer Institute of United States Public Health Service (R01 CA122443, P30 CA15083, P50 CA136393, and GAME-ON the NCI Cancer Post-GWAS Initiative U19 CA148112), the Fred C and Katherine B Andersen Foundation, the Mayo Foundation,

and the Ovarian Cancer Research Fund with support of the Smith family, in memory of Kathryn Sladek Smith. MoMaTEC received financial support from a Helse Vest Grant, the University of Bergen, Melzer Foundation, The Norwegian Cancer Society (Harald Andersens legat), The Research Council of Norway and Haukeland University Hospital. The Newcastle Endometrial Cancer Study (NECS) acknowledges contributions from the University of Newcastle, the NBN Children's Cancer Research Group, Ms. Jennie Thomas, and the Hunter Medical Research Institute. RENDOCAS was supported through the regional agreement on medical training and clinical research (ALF) between Stockholm County Council and Karolinska Institutet (numbers: 20110222, 20110483, 20110141 and DF07015), The Swedish Labor Market Insurance (number 100069), and The Swedish Cancer Society (number 11 0439). The Cancer Hormone Replacement Epidemiology in Sweden Study (CAHRES, formerly called The Singapore and Swedish Breast/Endometrial Cancer Study; SASBAC) was supported by funding from the Agency for Science, Technology and Research of Singapore (A*STAR), the US National Institutes of Health, and the Susan G. Komen Breast Cancer Foundation. The WHI program is funded by the National Heart, Lung, and Blood Institute, the US National Institutes of Health, and the US Department of Health and Human Services (HHSN268201100046C, HHSN268201100001C, HHSN268201100002C, HHSN268201100003C, HHSN268201100004C, and HHSN271201100004C). This work was also funded by NCI U19 CA148065-01.

The Nurses' Health Study (NHS) is supported by the NCI, NIH Grants Number UM1 CA186107, P01 CA087969, R01 CA49449, 1R01 CA134958, and 2R01 CA082838. We thank the participants and staff of the Nurses' Health Study for their valuable contributions as well as the following state cancer registries for their help: AL, AZ, AR, CA, CO, CT, DE, FL, GA, ID, IL, IN, IA, KY, LA, ME, MD, MA, MI, NE, NH, NJ, NY, NC, ND, OH, OK, OR, PA, RI, SC, TN, TX, VA, WA, and WY. We assume full responsibility for analyses and interpretation of these data. We also thank Channing Division of Network Medicine, Department of Medicine, Brigham and Women's Hospital, and Harvard Medical School. Finally, we also acknowledge Pati Soule and Hardeep Ranu for their laboratory assistance. The Connecticut Endometrial Cancer Study was supported by NCI, NIH Grant Number RO1CA98346. The Fred Hutchinson Cancer Research Center (FHCRC) is supported by NCI, NIH Grant Number NIH RO1 CA105212, RO1 CA 87538, RO1 CA75977, RO3 CA80636, NO1 HD23166, R35 CA39779, KO5 CA92002, and funds from the Fred Hutchinson Cancer Research Center. The Multiethnic Cohort Study (MEC) is supported by the NCI, NIH Grants Number CA54281, CA128008, and 2R01 CA082838. The California Teachers Study (CTS) is supported by NCI, NIH Grant Number 2R01 CA082838, R01 CA91019, and R01 CA77398, and contract 97-10500 from the California Breast Cancer Research Fund. The Polish Endometrial Cancer Study (PECS) is supported by the Intramural Research Program of the NCI. The Prostate, Lung, Colorectal, and Ovarian Cancer Screening Trial (PLCO) is supported by the Extramural and the Intramural Research Programs of the NCI.

The WHI program is funded by the National Heart, Lung, and Blood Institute, National Institutes of Health, U.S. Department of Health and Human Services through contracts, HHSN268201600018C, HHSN268201600001C, HHSN268201600002C, HHSN268201600003C, and HHSN268201600004C. This manuscript was prepared in collaboration with investigators of the WHI, and has been reviewed and/or approved by the Women's Health Initiative (WHI). WHI investigators are listed at https://www.whi.org/researchers/Documents%20%20Write%20a%20Paper/WHI%20Investigator%20Short%20List.pdf.

The Breast Cancer Association Consortium (BCAC) is funded by Cancer Research UK (C1287/A10118, C1287/A12014). The Ovarian Cancer Association Consortium (OCAC) is supported by a grant from the Ovarian Cancer Research Fund thanks to donations by the family and friends of Kathryn Sladek Smith (PPD/RPCI.07), and the UK National Institute for Health Research Biomedical Research Centres at the University of Cambridge. This research has been conducted using the UK Biobank Resource under applications 5122 and 9797. We gratefully acknowledge the TCGA endometrial cancer consortium for providing samples, tissues, data processing, and making data and results available. Additional funding for individual control groups is detailed in the Supplementary Information.

## Author contributions

T.A.O'M., D.J.T., A.B.S., and D.F.E. designed the study; T.A.O'M., D.J.T., D.M.G., and A.B.S. drafted the manuscript; T.A.O'M. and D.J.T. conducted statistical analyses and genotype imputation; T.A.O'M. and D.M.G. conducted bioinformatic analyses; T.A.O'M., J.F., and G.W.M. conducted eQTL analyses; D.M.G. and T.A.O'M. performed and analyzed ChIP-seq experiments; T.A.O'M. co-ordinated the iCOGS and OncoArray case genotyping, and associated data management; J.D., J.P.T., and K.M. co-ordinated quality control and data cleaning for the iCOGS and OncoArray control datasets; A.B.S. and T.A.O'M. co-ordinated the ANECS stage 1 genotyping; A.M.D. and C.S.H. co-ordinated the SEARCH stage 1 genotyping; I.T. and CHIBCHA funded and implemented the NSECG GWAS; I.T., L.M., M.G., A.J., and S.H. co-ordinated the National Study of Endometrial Cancer Genetics (NSECG), and collation of CORGI control GWAS data; A.B.S. and P.M.W. co-ordinated the Australian National Endometrial Cancer Study (ANECS); R.J.S., M.M.C.E., J.A., and E.G.H. co-ordinated collation of GWAS data for the Hunter Community Study; P.D.P.P., D.F.E., and M.S. co-ordinated Studies of Epidemiology and Risk Factors in Cancer Heredity (SEARCH); M.K.B. provided data

management support for BCAC; I.D.V., P.K., and M.M.C. co-ordinated E2C2 genotyping and analysis; F.D. and J.P. co-ordinated analysis of UK Biobank genotyping data; The following authors provided samples and/or phenotypic data: F.A., D.A., K.A., J.A., P.L.A., M.W.B., A.B., H.B., H.Br., L.B., D.D.B., B.B., J.C.-C., S.J.C., C.C., C.L.C., M.C., L.S.C., F.J. C., A.C., L.C., J.De., J.A.D., S.C.D., A.B.E., P.A.F., B.L.F., L.F., M.M.G., G.G.G., E.L.G., C. A.H., P.H., S.H., A.H., P.Hi., E.H., J.L.H., D.J.H., C.K., V.N.K., D.L., L.L.M., E.L., A.L., J. L., J.Lo., L.L., A.M.M., A.M., R.L.M., M.M., R.N., H.O., I.O., G.O., C.P., J.P., L.P., J.Pr., T. P., T.R.R., H.A.R., R.A.W.R., I.R., C.S., G.E.S., F.S., V.W.S., X.S., X.-O.S., M.C.S., A.J.S., E. T., J.T., C.T., C.V., D.V.D.B., A.V., Z.W., N.W., H.M.J.W., S.J.W., A.W., L.X., Y.-B.X., H. P.Y., and H.Y. All authors provided critical review of the manuscript.

### Additional information

**Competing interests:** The authors declare no competing interests.

Tracy A. O'Mara[1], Dylan M. Glubb[1], Frederic Amant[2], Daniela Annibali[2], Katie Ashton[3,4,5], John Attia[3,6], Paul L. Auer[7,8], Matthias W. Beckmann[9], Amanda Black[10], Manjeet K. Bolla[11], Hiltrud Brauch[12,13,14], Hermann Brenner[14,15,16], Louise Brinton[10], Daniel D. Buchanan[17,18,19,20], Barbara Burwinkel[21,22], Jenny Chang-Claude[23,24], Stephen J. Chanock[10], Chu Chen[25], Maxine M. Chen[26], Timothy H.T. Cheng[27], Christine L. Clarke[28], Mark Clendenning[17,20], Linda S. Cook[29,30], Fergus J. Couch[31], Angela Cox[32], Marta Crous-Bous[26,33], Kamila Czene[34], Felix Day[35], Joe Dennis[11], Jeroen Depreeuw[2,36,37], Jennifer Anne Doherty[38], Thilo Dörk[39], Sean C. Dowdy[40], Matthias Dürst[41], Arif B. Ekici[42], Peter A. Fasching[9,43], Brooke L. Fridley[44], Christine M. Friedenreich[30], Lin Fritschi[45], Jenny Fung[46], Montserrat García-Closas[10,47], Mia M. Gaudet[48], Graham G. Giles[18,49,50], Ellen L. Goode[51], Maggie Gorman[27], Christopher A. Haiman[52], Per Hall[34,53], Susan E. Hankison[33,54], Catherine S. Healey[55], Alexander Hein[9], Peter Hillemanns[39], Shirley Hodgson[56], Erling A. Hoivik[57,58], Elizabeth G. Holliday[3,6], John L. Hopper[18], David J. Hunter[26,59], Angela Jones[27], Camilla Krakstad[57,58], Vessela N. Kristensen[60,61,62], Diether Lambrechts[37,63], Loic Le Marchand[64], Xiaolin Liang[65], Annika Lindblom[66], Jolanta Lissowska[67], Jirong Long[68], Lingeng Lu[69], Anthony M. Magliocco[70], Lynn Martin[71], Mark McEvoy[6], Alfons Meindl[72], Kyriaki Michailidou[11,73], Roger L. Milne[18,49], Miriam Mints[74], Grant W. Montgomery[1,46], Rami Nassir[75], Håkan Olsson[76], Irene Orlow[65], Geoffrey Otton[77], Claire Palles[27], John R.B. Perry[35], Julian Peto[78], Loreall Pooler[52], Jennifer Prescott[33], Tony Proietto[77], Timothy R. Rebbeck[79,80], Harvey A. Risch[69], Peter A.W. Rogers[81], Matthias Rübner[82], Ingo Runnebaum[41], Carlotta Sacerdote[83,84], Gloria E. Sarto[85], Fredrick Schumacher[86], Rodney J. Scott[3,4,5,87], V. Wendy Setiawan[52], Mitul Shah[55], Xin Sheng[52], Xiao-Ou Shu[68], Melissa C. Southey[17,88], Anthony J. Swerdlow[89,90], Emma Tham[66,91], Jone Trovik[57,58], Constance Turman[26], Jonathan P. Tyrer[55], Celine Vachon[92], David VanDen Berg[52], Adriaan Vanderstichele[93], Zhaoming Wang[10], Penelope M. Webb[94], Nicolas Wentzensen[10], Henrica M.J. Werner[57,58], Stacey J. Winham[95], Alicja Wolk[96], Lucy Xia[52], Yong-Bing Xiang[97], Hannah P. Yang[10], Herbert Yu[64], Wei Zheng[68], Paul D.P. Pharoah[11,55], Alison M. Dunning[55], Peter Kraft[26,59], Immaculata De Vivo[26,33], Ian Tomlinson[27,71], Douglas F. Easton[11,55], Amanda B. Spurdle[1] & Deborah J. Thompson[11]

[1]Department of Genetics and Computational Biology, QIMR Berghofer Medical Research Institute, Brisbane 4006 QLD, Australia. [2]Department of Obstetrics and Gynecology, University Hospitals KU Leuven, University of Leuven, Division of Gynecologic Oncology, Leuven 3000, Belgium. [3]John Hunter Hospital, Hunter Medical Research Institute, Newcastle 2305 NSW, Australia. [4]University of Newcastle, Centre for Information Based Medicine, Callaghan 2308 NSW, Australia. [5]University of Newcastle, Discipline of Medical Genetics, School of Biomedical Sciences and Pharmacy, Faculty of Health, Callaghan 2308 NSW, Australia. [6]University of Newcastle, Centre for Clinical Epidemiology and Biostatistics, School of Medicine and Public Health, Callaghan 2308 NSW, Australia. [7]Cancer Prevention Program, Fred Hutchinson Cancer Research Center, Seattle 98109 WA,

USA. [8]University of Wisconsin-Milwaukee, Zilber School of Public Health, Milwaukee 53205 WI, USA. [9]Department of Gynecology and Obstetrics, University Hospital Erlangen, Friedrich-Alexander-University Erlangen-Nuremberg, Comprehensive Cancer Center ER-EMN, Erlangen 91054, Germany. [10]National Cancer Institute, Division of Cancer Epidemiology and Genetics, Bethesda 20892 MD, USA. [11]Department of Public Health and Primary Care, University of Cambridge, Centre for Cancer Genetic Epidemiology, Cambridge CB1 8RN, UK. [12]Dr. Margarete Fischer-Bosch-Institute of Clinical Pharmacology, Stuttgart 70376, Germany. [13]University of Tübingen, Tübingen 72074, Germany. [14]German Cancer Research Center (DKFZ), German Cancer Consortium (DKTK), Heidelberg 69120, Germany. [15]Division of Clinical Epidemiology and Aging Research, German Cancer Research Center (DKFZ), Heidelberg 69120, Germany. [16]Division of Preventive Oncology, German Cancer Research Center (DKFZ) and National Center for Tumor Diseases (NCT), Heidelberg 69120, Germany. [17]Department of Clinical Pathology, The University of Melbourne, Melbourne 3010 VIC, Australia. [18]The University of Melbourne, Centre for Epidemiology and Biostatistics, Melbourne School of Population and Global Health, Melbourne 3010 VIC, Australia. [19]Genetic Medicine and Family Cancer Clinic, Royal Melbourne Hospital, Parkville 3010 VIC, Australia. [20]Victorian Comprehensive Cancer Centre, University of Melbourne Centre for Cancer Research, Parkville 3010 VIC, Australia. [21]Department of Obstetrics and Gynecology, University of Heidelberg, Heidelberg 69120, Germany. [22]Molecular Epidemiology Group, C080, German Cancer Research Center (DKFZ), Heidelberg 69120, Germany. [23]Division of Cancer Epidemiology, German Cancer Research Center (DKFZ), Heidelberg 69120, Germany. [24]University Medical Center Hamburg-Eppendorf, Cancer Epidemiology, University Cancer Center Hamburg (UCCH), Hamburg 20246, Germany. [25]Epidemiology Program, Fred Hutchinson Cancer Research Center, Seattle 98109 WA, USA. [26]Department of Epidemiology, Harvard T.H. Chan School of Public Health, Boston 02115 MA, USA. [27]University of Oxford, Wellcome Trust Centre for Human Genetics and Oxford NIHR Biomedical Research Centre, Oxford OX3 7BN, UK. [28]University of Sydney, Westmead Institute for Medical Research, Sydney 2145 NSW, Australia. [29]University of New Mexico, University of New Mexico Health Sciences Center, Albuquerque 87131 NM, USA. [30]Department of Cancer Epidemiology and Prevention Research, Alberta Health Services, Calgary T2N 4N2 AB, Canada. [31]Department of Laboratory Medicine and Pathology, Mayo Clinic, Rochester 55905 MN, USA. [32]Department of Oncology and Metabolism, University of Sheffield, Sheffield Institute for Nucleic Acids (SInFoNiA), Sheffield S10 2TN, UK. [33]Department of Medicine, Harvard Medical School, Channing Division of Network Medicine, Brigham and Women's Hospital, Boston 02115 MA, USA. [34]Department of Medical Epidemiology and Biostatistics, Karolinska Institutet, Stockholm 171 65, Sweden. [35]University of Cambridge, MRC Epidemiology Unit, School of Clinical Medicine, Cambridge CB2 0QQ, UK. [36]VIB, Vesalius Research Center, Leuven 3000, Belgium. [37]Department of Human Genetics, University of Leuven, Laboratory for Translational Genetics, Leuven 3000, Belgium. [38]Cancer Research Huntsman Cancer Institute Department of Population Health Sciences, University of Utah, Salt Lake City 84112 UT, USA. [39]Gynaecology Research Unit, Hannover Medical School, Hannover 30625, Germany. [40]Department of Obstetrics and Gynecology, Mayo Clinic, Division of Gynecologic Oncology, Rochester 55905 MN, USA. [41]Department of Gynaecology, Jena University Hospital - Friedrich Schiller University, Jena 07743, Germany. [42]Friedrich-Alexander University Erlangen-Nuremberg, Comprehensive Cancer Center Erlangen-EMN, Institute of Human Genetics, University Hospital Erlangen, Erlangen 91054, Germany. [43]Department of Medicine, University of California at Los Angeles, David Geffen School of Medicine, Division of Hematology and Oncology, Los Angeles 90095 CA, USA. [44]Department of Biostatistics, Kansas University Medical Center, Kansas City 66160 KS, USA. [45]Curtin University, School of Public Health, Perth 6102 WA, Australia. [46]University of Queensland, Institute for Molecular Bioscience, Brisbane 4072 QLD, Australia. [47]Institute of Cancer Research, Division of Genetics and Epidemiology, London SM2 5NG, UK. [48]American Cancer Society, Epidemiology Research Program, Atlanta 30303 GA, USA. [49]Cancer Epidemiology and Intelligence Division, Cancer Council Victoria, Melbourne 3004 VIC, Australia. [50]Department of Epidemiology and Preventive Medicine, Monash University, Melbourne 3004 VIC, Australia. [51]Department of Health Science Research, Mayo Clinic, Division of Epidemiology, Rochester 55905 MN, USA. [52]Department of Preventive Medicine, University of Southern California, Keck School of Medicine, Los Angeles 90033 CA, USA. [53]Department of Oncology, South General Hospital, Stockholm 118 83, Sweden. [54]Department of Biostatistics and Epidemiology, University of Massachusetts, Amherst, Amherst 01003 MA, USA. [55]Department of Oncology, University of Cambridge, Centre for Cancer Genetic Epidemiology, Cambridge CB1 8RN, UK. [56]Department of Clinical Genetics, St George's, University of London, London SW17 0RE, UK. [57]Department of Clinical Science, University of Bergen, Centre for Cancer Biomarkers, Bergen 5020, Norway. [58]Department of Gynecology and Obstetrics, Haukeland University Hospital, Bergen 5021, Norway. [59]Program in Genetic Epidemiology and Statistical Genetics, Harvard T.H. Chan School of Public Health, Boston 02115 MA, USA. [60]Department of Cancer Genetics, Oslo University Hospital, Radiumhospitalet, Institute for Cancer Research, Oslo 0379, Norway. [61]University of Oslo, Institute of Clinical Medicine, Faculty of Medicine, Oslo 0450, Norway. [62]Department of Clinical Molecular Biology, University of Oslo, Oslo University Hospital, Oslo 0450, Norway. [63]VIB, VIB Center for Cancer Biology, Leuven 3001, Belgium. [64]Epidemiology Program, University of Hawaii Cancer Center, Honolulu 96813 HI, USA. [65]Department of Epidemiology and Biostatistics, Memorial Sloan-Kettering Cancer Center, New York 10065 NY, USA. [66]Department of Molecular Medicine and Surgery, Karolinska Institutet, Stockholm 171 76, Sweden. [67]Department of Cancer Epidemiology and Prevention, M. Sklodowska-Curie Cancer Center-Oncology Institute, Warsaw 02-034, Poland. [68]Department of Medicine, Vanderbilt University School of Medicine, Division of Epidemiology, Vanderbilt Epidemiology Center, Vanderbilt-Ingram Cancer Center, Nashville 37232 TN, USA. [69]Chronic Disease Epidemiology, Yale School of Public Health, New Haven 06510 CT, USA. [70]Department of Anatomic Pathology, Moffitt Cancer Center and Research Institute, Tampa 33612 FL, USA. [71]University of Birmingham, Institute of Cancer and Genomic Sciences, Birmingham B15 2TT, UK. [72]Department of Gynecology and Obstetrics, Ludwig-Maximilians University of Munich, Munich 80336, Germany. [73]Department of Electron Microscopy/Molecular Pathology, The Cyprus Institute of Neurology and Genetics, Nicosia, Cyprus. [74]Department of Women's and Children's Health, Karolinska Institutet, Stockholm 171 76, Sweden. [75]Department of Biochemistry and Molecular Medicine, University of California Davis, Davis 95817 CA, USA. [76]Department of Cancer Epidemiology, Clinical Sciences, Lund University, Lund 222 42, Sweden. [77]University of Newcastle, School of Medicine and Public Health, Callaghan 2308 NSW, Australia. [78]Department of Non-Communicable Disease Epidemiology, London School of Hygiene and Tropical Medicine, London WC1E 7HT, UK. [79]Harvard T.H. Chan School of Public Health, Boston 02115 MA, USA. [80]Dana-Farber Cancer Institute, Boston 02115 MA, USA. [81]Department of Obstetrics and Gynaecology, University of Melbourne, Royal Women's Hospital, Gynaecology Research Centre, Parkville 3052 VIC, Australia. [82]Department of Gynaecology and Obstetrics, Friedrich-Alexander University Erlangen-Nuremberg, Comprehensive Cancer Center Erlangen-EMN, University Hospital Erlangen, Erlangen 91054, Germany. [83]Center for Cancer Prevention (CPO-Peimonte), Turin 10126, Italy. [84]Human Genetics Foundation (HuGeF), Turino 10126, Italy. [85]Department of Obstetrics and Gynecology, University of Wisconsin, School of Medicine and Public Health, Madison 53715 WI, USA. [86]Department of Epidemiology and Biostatistics, Case Western Reserve University, Cleveland 44106 OH, USA. [87]John Hunter Hospital, Division of Molecular Medicine, Pathology North, Newcastle 2308 NSW, Australia. [88]Monash University, Precision Medicine, School of Clinical Sciences at Monash Health, Clayton 3168 VIC, Australia. [89]Division of Genetics and Epidemiology, The Institute of Cancer Research, London SM2 5NG, UK. [90]Division of Breast Cancer Research, The Institute of Cancer Research, London SW7 3RP, UK. [91]Karolinska Institutet, Clinical Genetics, Stockholm 171 76, Sweden. [92]Department of Health Sciences Research, Mayo Clinic, Rochester 55905 MN, USA. [93]Department of Obstetrics and Gynaecology, University Hospitals Leuven, Division of Gynecologic Oncology, Leuven Cancer Institute,

Leuven 3000, Belgium. [94]Department of Population Health, QIMR Berghofer Medical Research Institute, Brisbane 4006 QLD, Australia. [95]Department of Health Science Research, Mayo Clinic, Division of Biomedical Statistics and Informatics, Rochester 55905 MN, USA. [96]Department of Environmental Medicine, Karolinska Institutet, Division of Nutritional Epidemiology, Stockholm 171 77, Sweden. [97]Department of Epidemiology, Shanghai Cancer Institute, Renji Hospital, Shanghai Jiaotong University School of Medicine, State Key Laboratory of Oncogene and Related Genes, Shanghai, China. These authors contributed equally: Amanda B. Spurdle, Deborah J. Thompson.

