## [Peer Review File · Nature Communications]

Reviewers' comments:

Reviewer #1 (Remarks to the Author):

This is an important contribution to the literature and greatly increases the number of participants who have been genotyped and analyzed for endometrial cancer. Generally the manuscript is well-written. However, there are some components I find confusing or incomplete. The organization of the study is clearly presented. The statistical approaches are clearly presented (I have a minor request for clarification). Analyses seem comprehensive and appropriate.

Perhaps the most important point is the finding for rs9639594 seems very poorly supported by the available data. In particular, in the regional plot this is the only SNP that showed substantial evidence of an effect and the I² value is 41% indicating that the evidence for it is being provided by only a few studies. Also, this SNP had the lowest Info. To evaluate further if this signal is real or some sort of artifact a study by study table of the OR and P-values for each study should be presented in an appendix. Finally given uncertainty about whether the finding is accurate perhaps imputation with the Michigan imputation server for the data that are available on hand should be performed to further improve the imputation accuracy for this snp and then its result could be checked again to evaluate if it remains genome-wide significant.

In table 1, the BFDP for rs2498796 does not seem consistent with the other results with similar p-value so I wonder if the 98% BFDP result is accurate. Please check.

In the online methods there are some useful presentations of methods used for calculating FRR and heritability. However, there is also the sentence: The frailty-scale heritability (as opposed to the observed-scale heritability) was obtained by replacing the total sample with an effective sample size N_j for SNP j thus: ... with the equation for N_j . However, the manuscript does not explain how N_j is subsequently used, so I found this section confusing. Please clarify.

The authors did not cite the support for the Oncoarray correctly : in particular they failed to cite the grants that supported the genotyping, which they included. The following sentence should be inserted somewhere: CIDR genotyping for the Oncoarray was conducted under contract 268201200008I to Johns Hopkins University and through grant X01HG007491. The Oncoarray grant had multiple PIs and if you need to cite a specific person for it you can cite Tom Sellers.

In supplementary table 1 column p row 27 there is a stray number, please delete.

I was thinking it might be useful to present the citations for the functional assays in supplemental table 3 but this is not a required change as long as the assays are cited in the paper somewhere else (its just a bit difficult to go back and forth between the table and the paper).

Reviewer #2 (Remarks to the Author):

This study reports the discovery of ten new loci for endometrial cancer, including one locus specific for the endometrioid subtype from a meta-analysis of 17 GWAS including ~12,000 cases. It is the largest GWAS to date of the cancer with 3,613 subjects that had not been included in any GWAS to date. Overall, it is a well conducted study and adds insight into the genetic architecture of endometrial cancer, but some of the methods are a bit unclear and I would like to see some additional support for the loci identified. The authors do not provide any replication for their results, which is not uncommon in the era of GWAS meta-analyses. However, since many of their loci are imputed, it would be nice to have some technical validation of loci with second genotyping technology to ensure readers that findings are not the result of a platform artifact. Additional comments are below:

- 1) The description of the GWAS and how they were analyzed in the methods is a bit hard to follow. The authors should provide a figure showing the studies included, sample sizes, genotyping platform and how they were combined for the meta-analysis. Some studies, such as E2C2 and WHI, had subjects genotyped on different platforms, but it is unclear if they were imputed separately or together. A table showing the QC and imputation steps for each study would help to clarify the methods. The authors also mention that no analyses were undertaken to identify duplicates or related samples between studies. However, the authors could have done some pairwise meta-analyses between studies to see if there was inflation, suggesting some shared subjects.
- 2) The authors estimate the proportion of familial relative risk explained by the identified SNPs, but it is unclear what beta estimates they used to make this calculation, both for the previously discovered loci as well as the new loci. The beta estimates for the new loci are likely inflated due to winner's curse. The beta estimates from the old loci may also be inflated if the scans used in their initial discovery were used in generating the betas.
- 3) The authors provide the info score from the oncoarray, but not from any of the other GWAS used in this meta-analysis. The authors should provide the info scores from each GWAS for each SNP or at least the range of info scores for each SNP, so that the reader can get a sense of the quality of imputation across studies. It would also be helpful to see the effect allele frequencies for cases and controls in each GWAS.
- 4) The authors report one locus specific for the endometrioid subtype; however, on the locuszoom plot, it appears to be a singleton, despite having a frequency of 0.21. This is a bit worrisome. The authors should provide additional replication, technical validation, or evidence to assure the reader that it is not a false positive.
- 5) The authors report enrichment for variants associated with genes expressed in specific tissues, but it is not clear how the authors determined enrichment or the significance level used, given that multiple tissues were tested. Also, it would be helpful if the authors provided the betas/p-values for the eqtl associations, so that the reader could get a sense of the strength of the evidence for each locus. If the authors look at the previously report loci as well as the new loci, do any candidate pathways emerge, providing a unifying theme?
- 6) The authors report greater overlap for these epigenomic features with the ccrSNPs

related to endometrial cancer compared to endometriosis or schizophrenia, but it is unclear how this comparison was made and p-value calculated and whether LD structure and proximity to genes were taken into account.

7) The authors expand up their previous Mendelian randomization, but I wonder if this would be better off in a separate paper, where more detailed analyses could be presented.

8) Minor: Table 1 and 2 should include the sample sizes for the histologies. Some references are missing in the results section for the methods used.

Reviewer #3 (Remarks to the Author):

This manuscript describes a meta-GWAS analysis of 12,906 endometrial cancer cases and 108,979 country matched controls identifying 10 new additional risk loci for endometrial cancer. The authors state that the identification of these loci contributes an additional 2.8% for a total of 6.9%, thus explaining about a quarter of the portion of the FRR attributable to common SNPs.

The results add additional risk loci to those they have already reported. The authors breakdown the risk loci between endometrioid and non-endometrioid histologies and interestingly show only one locus that demonstrates a significant difference, in contrast to 4/10 of the previously described loci. In addition, the expanded analysis strengthened the association between BMI and the protective effect of later menarche. Also, of note is the presence of several genes within the loci that are known to be altered in endometrial carcinoma (e.g., NF1).

These studies are of considerable interest to the field as they provide possible targets for future studies of endometrial cancer pathogenesis. Like all GWAS studies they are descriptive and require concerted, future effort to determine the significance of specific genes within the identified loci. These studies also found that most of the newly identified risk loci are common between endometrioid and non-endometrioid histology. Unfortunately, the significance of this finding is unclear in light of the TCGA study suggesting that Grade 3 endometrioid tumors share some genetic similarities to serous carcinoma, the most common non-endometrioid histology. In addition, some epidemiologic studies have suggested overlap in the risk factors between serous and Grade 3 endometrioid carcinoma. In addition, other non-endometrioid histologies are likely to be genetically heterogenous. If this more granular detail is available on the cohort of endometrial cancers, and the numbers are sufficient for meaningful data, it might provide additional novel insights into the risk factors for the various types of endometrial cancer.

Reviewer #1:

Perhaps the most important point is the finding for rs9639594 seems very poorly supported by the available data. In particular, in the regional plot this is the only SNP that showed substantial evidence of an effect and the r^2 value is 41% indicating that the evidence for it is being provided by only a few studies. Also, this SNP had the lowest Info. To evaluate further if this signal is real or some sort of artifact a study by study table of the OR and P-values for each study should be presented in an appendix. Finally given uncertainty about whether the finding is accurate perhaps imputation with the Michigan imputation server for the data that are available on hand should be performed to further improve the imputation accuracy for this snp and then its result could be checked again to evaluate if it remains genome-wide significant.

Since submission, we have been able to access genotypes imputed using the Haplotype Reference Consortium reference panel for one subset of the cases and controls. The results including this updated imputation are extremely similar to those in the manuscript for nine of the novel SNPs in Table 2, but rs9639594 failed to reach genome-wide significance, either overall or in the Type 1 histology group (see Table below). We have therefore rewritten our results, noting that it is a single, imputed SNP and needs to be explored further in future studies. We have removed this SNP from our list of discovered loci, familial relative risk calculations and functional annotations. The result for this SNP is still presented in Supplementary Table 2, but now accompanied by a footnote explaining why it was not presented as a GWAS-significant locus in the paper.

Given the preliminary and limited extent of the HRC imputation currently available to us, and the very minor changes to the other nine loci, we prefer not to include this information in the manuscript.

	1000 Genomes imputation for all studies		HRC imputation for OncoArray study (1000 Genomes for all other studies)		
SNP	Info score for OncoArray studies	All histologies	Info score for OncoArray studies	All histologies	Type 1 Histology
rs113998067	0.90	1.23 (1.14, 1.32) P=3.6E-8	0.93	1.23 (1.14-1.32) p=2.02e-8	
rs148261157	0.88	1.26 (1.16, 1.36) P=3.4E-8	0.94	1.25 (1.16-1.36) p=1.58e-8	
rs9639594	0.85	1.08 (1.04, 1.13) P=6.5E-5	0.91	1.07 (1.03-1.12) p=3.51e-4	1.12 (1.07-1.17) p=4.46e-7
rs1679014	Genotyped	1.18 (1.12, 1.25) P=6.4E-9	Genotyped	1.18 (1.12-1.25) p=5.22e-9	
rs10835920	0.99	1.09 (1.06, 1.13) P=1.3E-8	1.00	1.09 (1.06-1.13) p=1.15e-8	
rs9668337	0.99	1.11 (1.08, 1.15) P=1.1E-9	1.00	1.11 (1.07-1.15) p=1.38e-9	
rs3184504	Genotyped	1.10 (1.07, 1.14) P=1.1E-10	Genotyped	1.10 (1.07-1.14) p=1.34e-10	
rs10850382	Genotyped	1.10 (1.07, 1.14) P=3.5E-9	Genotyped	1.10 (1.07-1.14) p=3.57e-9	
rs1129506	Genotyped	1.10 (1.06, 1.13) P=4.3E-8	Genotyped	1.10 (1.06-1.13) p=4.42e-8	
rs882380	0.99	1.10 (1.06, 1.13) P=4.7E-9	1.00	1.10 (1.06-1.13) p=5.10e-9	

In table 1, the BFDP for rs2498796 does not seem consistent with the other results with similar p-value so I wonder if the 98% BFDP result is accurate. Please check.

Yes, this is correct and is a reflection of the fact that this previously reported SNP was not replicated in the current study.

In the online methods there are some useful presentations of methods used for calculating FRR and heritability. However, there is also the sentence: The frailty-scale heritability (as opposed to the observed-scale heritability) was obtained by replacing the total sample with an effective sample size N_j for SNP j thus... with the equation for N_j . However, the manuscript does not explain how N_j is subsequently used, so I found this section confusing. Please clarify.

The LD Score Regression program requires a sample size, N , for each SNP. We replaced the total N for each SNP with the effective sample size, calculated as shown, which effectively weights SNPs according to their frequency and the variance of their effect estimates. We have added these details to the methods.

The authors did not cite the support for the Oncoarray correctly : in particular they failed to cite the grants that supported the genotyping, which they included. The following sentence should be inserted somewhere: CIDR genotyping for the Oncoarray was conducted under contract 2682012000081 to Johns Hopkins University and through grant X01HG007491. The Oncoarray grant had multiple PIs and if you need to cite a specific person for it you can cite Tom Sellers.

We have conferred once again with our co-authors who represent OCAC (in addition to ECAC) regarding the acknowledgements section relevant to OCAC. We have altered the relevant text to read:

Genotyping of ECAC cases was performed with the generous assistance of the Ovarian Cancer Association Consortium (OCAC). We particularly thank the efforts of Cathy Phelan. The OCAC OncoArray genotyping project was funded through grants from the US National Institutes of Health (CA1X01HG007491-01 (Christopher I. Amos), U19-CA148112 (Thomas A. Sellers), R01-CA149429 (Catherine M. Phelan) and R01-CA058598 (Marc T. Goodman)); Canadian Institutes of Health Research (MOP-86727 (Linda E. Kelemen)); and the Ovarian Cancer Research Fund (Andrew Berchuck). CIDR genotyping for the Oncoarray was conducted under contract 2682012000081.

In supplementary table 1 column p row 27 there is a stray number, please delete.

This has been amended.

I was thinking it might be useful to present the citations for the functional assays in supplemental table 3 but this is not a required change as long as the assays are cited in the paper somewhere else (its just a bit difficult to go back and forth between the table and the paper).

These citations have been added as footnotes to the table.

Reviewer #2:

This study reports the discovery of ten new loci for endometrial cancer, including one locus specific for the endometrioid subtype from a meta-analysis of 17 GWAS including ~12,000 cases. It is the largest GWAS to date of the cancer with 3,613 subjects that had not been included in any GWAS to date. Overall, it is a well conducted study and adds insight into the genetic architecture of endometrial cancer, but some of the methods are a bit unclear and I would like to see some additional support for the loci identified. The authors do not provide any replication for their results, which is not uncommon in the era of GWAS meta-analyses. However, since many of their loci are imputed, it would be nice to have some technical validation of loci with second genotyping technology to ensure readers that findings are not the result of a platform artifact. Additional comments are below:

The SNP which had been least-well imputed (rs9639594, 7p14.3) has been removed from the paper (please see response to Reviewer 1). Imputation information scores for each of the nine genotyping projects have been added to Supplementary Table 2; across nine genotyping projects, the

information scores were >0.8 for eight of the nine variants now presented as representing novel risk loci. Moreover, for the only variant (rs148261157) which had an information score <0.8 in the iCOGS project, the average information score from all nine genotyping studies was 0.89. Considering imputation for these nine SNPs in the OncoArray case-control study, which comprises the largest component of the meta-analysis (36% of cases), four had been directly genotyped in the OncoArray study, three had imputation information scores of 0.99, and two had imputation information scores ≥ 0.88 . In view of this, and given the consistency of the results across the different studies and different platforms, we hope that the editor will agree that these findings are extremely unlikely to be the result of a platform artefact.

1) The description of the GWAS and how they were analyzed in the methods is a bit hard to follow. The authors should provide a figure showing the studies included, sample sizes, genotyping platform and how they were combined for the meta-analysis. Some studies, such as E2C2 and WHI, had subjects genotyped on different platforms, but it is unclear if they were imputed separately or together. A table showing the QC and imputation steps for each study would help to clarify the methods.

The information requested about studies, sample sizes and platforms is all detailed in ST1, and the details regarding the meta-analysis are described in the Methods. For E2C2 and for WHI, samples genotyped using separate platforms were imputed separately – we have now clarified this in the Methods.

The authors also mention that no analyses were undertaken to identify duplicates or related samples between studies. However, the authors could have done some pairwise meta-analyses between studies to see if there was inflation, suggesting some shared subjects.

Pairwise comparisons to identify duplicates/relatives were made among all samples genotyped as part of the OncoArray, iCOGS or ANECS/SEARCH/NSECG GWAS genotyping projects, as described in the Methods. These studies together account for over 70% of the cases. Of the three remaining studies (WHI, E2C2, UK Biobank), it seems very unlikely that there would have been any meaningful level of sample overlap. The sampling frame for the US studies WHI and E2C2 is not expected to overlap with the Mayo study, which provided a relatively small number of cases to ECAC. Although a small amount of overlap is potentially possible between UK Biobank and the other UK studies, the UK SEARCH study is based in the East Anglia region of the UK, which was not covered by the UK Biobank recruitment process. (Note that the UK Biobank expressly forbids researchers from attempting to match genetic data to genotypes from other studies.) The lack of evidence of inflation in test statistics is a further reassurance that sample overlap/relatedness was not a problem.

2) The authors estimate the proportion of familial relative risk explained by the identified SNPs, but it is unclear what beta estimates they used to make this calculation, both for the previously discovered loci as well as the new loci. The beta estimates for the new loci are like inflated due to winner's curse. The beta estimates from the old loci may also be inflated if the scans used in their initial discovery were used in generating the betas.

The logORs used to estimate the proportion of the FRR were those from this study. For the previously published SNPs these are slightly attenuated compared to the original reports. We acknowledge that the betas for the new SNPs (and to a lesser extent, the published SNPs) are susceptible to winner's curse, and hence this proportion is possibly a modest overestimate. We clarified this in the Methods ("The effect estimates used were those estimated in the current study, both for the new loci and for the loci replicated from previous studies.") and in the main text ("The common endometrial cancer variants identified to date together explain up to 6.8% of the FRR, including 2.7% contributed by the nine additional variants reported here; this may be an overestimate, given that the ORs for the new loci likely include some upwards bias (the so-called winner's curse)").

3) *The authors provide the info score from the oncoarray, but not from any of the other GWAS used in this meta-analysis. The authors should provide the info scores from each GWAS for each SNP or at least the range of info scores for each SNP, so that the reader can get a sense of the quality of imputation across studies. It would also be helpful to see the effect allele frequencies for cases and controls in each GWAS.*

Imputation info scores and effect-allele frequencies for each of the nine genotyping projects have been added to ST2. Allele frequencies are not available to us for the E2C2 study.

4) *The authors report one locus specific for the endometrioid subtype; however, on the locuszoom plot, it appears to be a singleton, despite having a frequency of 0.21. This is a bit worrisome. The authors should provide additional replication, technical validation, or evidence to assure the reader that it is not a false positive.*

Following further investigations, this SNP has been removed from the paper - please see responses to Reviewer 1 for details.

5) *The authors report enrichment for variants associated with genes expressed in specific tissues, but it is not clear how the authors determined enrichment or the significance level used, given that multiple tissues were tested.*

A dashed line has been added to Supp Figure 4 to show the threshold for significance with a FDR of 5% and a line has been added to the text to clarify this ("LD score regression analyses using eQTL results from GTEx¹¹ showed that endometrial cancer heritability exhibited the strongest evidence for enrichment for variants associated with genes specifically expressed in vaginal and uterine tissue, in line with prior assumptions, although none of the tissue-specific enrichments were significant after adjusting for the number of tissues tested (**Supplementary figure 4**)"). The figure legend has also been changed.

Also, it would be helpful if the authors provided the betas/p-values for the eqtl associations, so that the reader could get a sense of the strength of the evidence for each locus. If the authors look at the previously report loci as well as the new loci, do any candidate pathways emerge, providing a unifying theme?

Summary statistics for reported eQTL associations are now included in a new Supplementary Table (ST5).

Candidate pathway analysis using Ingenuity Pathway Analysis tool has been performed, including candidate causal genes from all known endometrial cancer risk loci. Analyses revealed a network, containing 18 of the 25 candidate causal genes, which was enriched for relevant pathways such as endometrial cancer signalling, adipogenesis, Wnt/ β -catenin signaling, estrogen-mediated S-phase entry, p53 signaling and PI3K/AKT signaling. These results have been added to the text.

6) *The authors report greater overlap for these epigenomic features with the ccrSNPs related to endometrial cancer compared to endometriosis or schizophrenia, but it is unclear how this comparison was made and p-value calculated and whether LD structure and proximity to genes were taken into account.*

We would like to stress that this is not an enrichment analysis, where it is important to ensure that the (null) comparison region has a similar LD structure and that proximity to genes are correctly accounted for. In this paper we were performing a comparison of the number of candidate causal risk SNPs (ccrSNPs) which intersected with functional elements mapped in cells/tissues relevant for endometrial cancer (endometrial cancer cell lines, uterine and vaginal tissue). The ccrSNPs for all three diseases were identified by their statistical probability of being functional variants and were then intersected with the epigenomic data. A Fisher's Exact test was performed to compare the difference in frequency of ccrSNPs which overlap these epigenomic features between endometrial cancer and either endometriosis or schizophrenia and the P-value provided in the text. The finding that 73% of the ccrSNPs for endometrial cancer and only 51% or endometriosis ccrSNPs or 40% of

schizophrenia ccrSNPs intersected with epigenomic features from endometrial cells provides good evidence that we are using the appropriate cells/tissues for functional annotation.

7) *The authors expand up their previous Mendelian randomization, but I wonder if this would be better off in a separate paper, where more detailed analyses could be presented.*

Since these Mendelian randomisation analyses are, in the main, updates to previously analysis, we feel that they do not justify separate papers, but rather provide examples of the value of an enlarged endometrial cancer GWAS as a resource for future studies of causality.

8) *Minor: Table 1 and 2 should include the sample sizes for the histologies.*

We have added this information to Table 1 and 2.

Some references are missing in the results section for the methods used.

We have added references for the software packages for which this was missing.

Reviewer #3 (Remarks to the Author):

This manuscript describes a meta-GWAS analysis of 12,906 endometrial cancer cases and 108,979 country matched controls identifying 10 new additional risk loci for endometrial cancer. The authors state that the identification of these loci contributes an additional 2.8% for a total of 6.9%, thus explaining about a quarter of the portion of the FRR attributable to common SNPs.

The results add additional risk loci to those they have already reported. The authors breakdown the risk loci between endometrioid and non-endometrioid histologies and interestingly show only one locus that demonstrates a significant difference, in contrast to 4/10 of the previously described loci. In addition, the expanded analysis strengthened the association between BMI and the protective effect of later menarche. Also, of note is the presence of several genes within the loci that are known to be altered in endometrial carcinoma (e.g., NF1).

These studies are of considerable interest to the field as they provide possible targets for future studies of endometrial cancer pathogenesis. Like all GWAS studies they are descriptive and require concerted, future effort to determine the significance of specific genes within the identified loci.

These studies also found that most of the newly identified risk loci are common between endometrioid and non-endometrioid histology. Unfortunately, the significance of this finding is unclear in light of the TCGA study suggesting that Grade 3 endometrioid tumors share some genetic similarities to serous carcinoma, the most common non-endometrioid histology. In addition, some epidemiologic studies have suggested overlap in the risk factors between serous and Grade 3 endometrioid carcinoma. In addition, other non-endometrioid histologies are likely to be genetically heterogeneous. If this more granular detail is available on the cohort of endometrial cancers, and the numbers are sufficient for meaningful data, it might provide additional novel insights into the risk factors for the various types of endometrial cancer.

We appreciate the comments of the reviewer. Unfortunately information on grade was limited for some case series in ECAC, and thus we are not currently able to address this issue. Whilst we did attempt to look for associations between SNPs and specific non-endometrioid histologies (serous, clear-cell, mucinous and carcinosarcoma) the numbers were too small to allow for meaningful results (ST1 and ST2). We have added an acknowledgement of this limitation to the text (“No SNP reached genome-wide significance in an analysis restricted to the 1,230 non-endometrioid cases (Figure 1d) or in separate analyses of carcinosarcomas, serous, clear cell or mucinous carcinomas, for which statistical power is very limited (ST2, Supplementary Figure 2).”).

REVIEWERS' COMMENTS:

Reviewer #1 (Remarks to the Author):

The authors have responded fully to all of my suggestions and those of the other reviewers. This is an important and well written paper.

Reviewer #2 (Remarks to the Author):

The authors have addressed my concerns.